# Synthesis of α-Aminophosphonates and Related Derivatives; The Last Decade of the Kabachnik–Fields Reaction

**DOI:** 10.3390/molecules26092511

**Published:** 2021-04-25

**Authors:** Petra R. Varga, György Keglevich

**Affiliations:** Department of Organic Chemistry and Technology, Budapest University of Technology and Economics, 1521 Budapest, Hungary; varga.petra.regina@vbk.bme.hu

**Keywords:** Kabachnik–Fields reaction, phospha-Mannich reaction, α-aminophosphonates, α-aminophosphine oxides, bis derivatives, catalysts, green methods, aza-Pudovik reaction

## Abstract

The Kabachnik–Fields reaction, comprising the condensation of an amine, oxo compound and a P-reagent (generally a >P(O)H species or trialkyl phosphite), still attracts interest due to the challenging synthetic procedures and the potential biological activity of the resulting α-aminophosphonic derivatives. Following the success of the first part (*Molecules* 2012, 17, 12821), here we summarize the synthetic developments in this field accumulated in the last decade. The procedures compiled include catalytic accomplishments as well as catalyst-free and/or solvent-free “greener” protocols. The products embrace α-aminophosphonates, α-aminophosphinates, and α-aminophosphine oxides along with different bis derivatives from the double phospha-Mannich approach. The newer developments of the aza-Pudovik reactions are also included.

## 1. Introduction

The Kabachnik–Fields reaction involves the condensation of a primary or secondary amine, an oxo compound such as an aldehyde or ketone, and a >P(O)H-containing reagent, which is in most cases a dialkyl phosphite, but may also be an alkyl-*H*-phosphinate or a secondary phosphine oxide, to result in the formation of α-aminophosphonates, α-aminophosphinates, and α-aminophosphine oxides, respectively [1,2,3,4,5]. The classical version of the “phospha-Mannich” reaction was discovered independently by Kabachnik and Fields more than sixty years ago [6,7].

The α-aminophosphonic and α-aminophosphinic derivatives incorporating an N–C–P moiety are still a focus due to their real or potential biological activity. The acid derivatives of the species under discussion may be regarded as the analogues of their natural counterparts, carboxylic acids. As such, as a consequence of their different properties (tetrahedral P vs. planar C, different acidity, and steric bulk) they are recognized by receptors and enzymes as false substrates/inhibitors [8,9,10,11,12]. The bioactivity realized in this way may be manifested in applications as agrochemicals and medicines.

The mechanism of the phospha-Mannich reaction depends on the nature of the substrates applied. It has been said that the condensation may proceed via an imine (Schiff base) or α-hydroxyphosphonate intermediate (Scheme 1/route “A” and “B” respectively) [13,14,15].

The number of publications describing different variations of the Kabachnik–Fields reaction exceeds 450 papers. The usual phospha-Mannich protocol includes the condensation of equimolar quantities of the three components in organic solvents and the use of various catalysts, comprising in most cases Lewis and Brönsted acids. A wide range of catalysts have been described, such as metal perchlorates; Amberlysts; succinic-, sulfonic-, and oxalic- acids; zinc, iron, and niobium salt; lanthanide triflates; boron trifluoride etherate; titanium dioxide; etc. [16]. However, it has been found that catalyst-free methods may also be appropriate, especially under solvent-free and/or microwave-assisted conditions [17,18,19,20,21,22,23]. The latter protocols represent green chemical approaches that are the focus of this article.

In this review, we summarize the new developments of the Kabachnik–Fields reaction accumulated in the last decade.

## 2. Kabachnik–Fields Reactions with Dialkyl Phosphites, Alkyl *H*-Phosphinates, and Secondary Phosphine Oxides

### 2.1. Metal-Catalyzed Kabachnik–Fields Reactions

In the first block, metal-catalyzed condensations are summarized. Applying (tetra-tert-buthylphthalocyaninato)aluminum chloride, a series of amines, acetophenone, and diethyl phosphite were condensed to afford the corresponding α-aminophosphonates (**1**) in acceptable yields (Scheme 2(1)). The reaction of benzylamine, indanone (**2**), and ethyl phenyl-*H*-phosphinate led to the respective phosphinate (**3**) (Scheme 2(2)) [24]. 

The three-component condensation of aniline and benzaldehyde derivatives with dialkyl phosphite (or with diphenyl phosphite) was performed in the presence of a cyclopentadienyl ruthenium(II) complex at 80 °C in a solvent-free manner. The α-aminophosphonates (**4**) were obtained in good to excellent yields (Scheme 3) [25].

Potentially biologically active α-aminophosphonates (**6**) were synthesized from quinazolinone-based hydrazides (**5**), aromatic aldehydes, and diphenyl phosphite using ZnCl_2_/PPh_3_ as the catalyst at room temperature (Scheme 4) [26]. The yields fell in the range of 75–84%. 

A similar ZnCl_2_·choline-chloride-catalyzed transformation of mostly aromatic amines, benzaldehyde derivatives, and diethyl phosphite led to α-amino-benzylphosphonates (**7**) in yields of 70–96% (Scheme 5) [27].

Zn di[bis(trifluoromethylsulfonyl)imide] (**11**) was also described as a catalyst in a similar condensation, and the product (**8**) was dearylated, suprisingly, by NBS; the intermediate (**9**) so formed was hydrolyzed (**10**) (Scheme 6) [28,29]. Application of an optically active additive (**12**, pybim) allowed an enantioselective synthesis.

Zinc triflate and iron triflate also proved to be efficient catalysts in the three-component condensation when different substituted amines or an oxadiazole-related acid hydrazide (**14**) were reacted with benzaldehyde and diethyl phosphite. These reactions took place under mild conditions, mostly in good yields (Scheme 7 and Scheme 8) [30,31].

The indium(III)-complex-catalyzed (**17, 18**) methods allowed the efficient condensation of various aldehydes, amines, and >P(O)H reagents under neat conditions at room temperature to give the corresponding α-aminophosphonates or α-aminophosphine oxides (**16**) (Scheme 9) [32]. Theyields were 86–98%. The need for special indium catalysts presents a disadvantage.

Nickel chloride and niobium chloride were efficient catalysts in the reaction of aniline derivatives, substituted benzaldehydes, and diethyl phosphite at a temperature of 82 °C (Scheme 10 and Scheme 11) [33,34]. The use of these catalysts gave the corresponding products (**19** and **4**) in similar yields.

A series of α-aminophosphonate derivatives (**21**) incorporating an uracil moiety was synthesized using Mg(ClO_4_)_2_ as the catalyst and acetonitrile as the solvent at 80 °C (Scheme 12) [35]. The products showed potential herbicidal activity.

A series of substituted diethyl α-phenylamino-benzylphosphonates (**4** (R^3^ = Et)) was prepared using cerium chloride or cerium oxide from aniline derivatives, substituted benzaldehydes, and diethyl phosphite under mild and solvent-free conditions (Scheme 13 and Scheme 14) [36,37]. The yields were 87–95%. 

Cu/Au and Gd oxide nanocatalysts allowed the efficient condensation of different amines, benzaldehyde derivatives, and dimethyl phosphite under conventional heating or under microwave (MW) irradiation (Scheme 15 and Scheme 16) [38,39]. The use of these catalysts gave the corresponding products (**22** and **24**) in similar yields.

Dehydroascorbic acid (DHAA)-capped magnetite nanoparticles were successfully applied in the phospha-Mannich condensation of aromatic amines and aldehydes with dimethyl phosphite (Scheme 17) [40].

A deep eutectic solvent (DES) comprising ZrOCl_2_·8H_2_O and urea in a 1:5 ratio made possible the efficient condensation of aryl-(heteroaryl)aldehydes, aniline derivatives and dimethyl phosphite at room temperature. The role of the DES was to serve as a reaction medium and as a catalyst (Scheme 18) [41].

In the above discussion, metal-catalyzed reactions are summarized. The application of metal catalysts makes possible efficient condensations, most of which occur at room temperature; however, these methods, especially the ones applying Zn or Ni promoters, cannot be considered environmentally friendly. At the same time, MW-assisted and solvent-free approaches may be considered as green techniques.

### 2.2. Acid-Catalyzed Kabachnik–Fields Reactions

The Kabachnik–Fields condensations outlined in the next section are promoted by acidic catalysts. The reaction of aniline derivatives, substituted benzaldehydes, and dimethyl phosphite was performed using a bifunctional acid–base catalyst (IRMOF-3, where MOF is a metal organic framework: Zn_4_O(H_2_N-TA)_3_ prepared from 2-aminoterephthalic acid (H_2_ATA) and Zn(NO_3_)_2_·6H_2_O) (Scheme 19) [42]. The other option was to use a sulfated polyborate as the catalyst (Scheme 20) [43]. When applying IRMOF-3 and sulfated polyborate as the catalyst, the corresponding products were formed in similar yields.

An efficient method was developed for the preparation of α-aminophosphonates (**26**), which involved applying phenylboronic acid as the catalyst under solvent-free conditions at 50 °C (Scheme 21) [44].

Phenylphosphonic acid was found to be an efficient reusable heterogeneous catalyst in the three-component phospha-Mannich reaction of benzylamine, aldehydes/ketones, and dimethyl phosphite (Scheme 22) [45].

The condensation of aminophenols, benzaldehyde derivatives, and dimethyl phosphite was performed in an aqueous medium containing oxalic acid as the catalyst at 90 °C. No yields were achieved (Scheme 23) [46].

The latter approach provided the α-aminophosphonates in almost quantitative yields. 

In another, more general example, a highly efficient biodegradable supramolecular polymer-supported catalyst was applied (Scheme 24) [47]. The catalyst represents a green component.

The readily available paratoluenesulfonic acid (PTSA) was found to be a suitable catalyst in a series of phospha-Mannich reactions. The thoroughly investigated model reaction is shown in Scheme 25 [48].

PTSA was also used in the condensation of different amines, formaldehyde, and secondary phosphine oxides in toluene at the boiling point (Scheme 26) [49]. The yields of products **29** and **30** were ≥90%.

New α-aminophosphine oxides and α-aminophosphonates (**32**/**33**) were made available by the PTSA-catalyzed reaction of amines containing an acetal group (**31**), paraformaldehyde, and >P(O)H reagents (Scheme 27 and Scheme 28) [50,51]. No yields were achieved for the second series of reactions.

Solvent-free realizations may be typically associated with MW irradiation. This was the case with the polystyrene-supported PTSA-catalyzed condensation of 2-aminofluorene (**34**), a number of aldehydes, and dimethyl phosphite (Scheme 29) [52].

β-Cyclodextrin-supported sulfonic acid was also used as an efficient and reusable heterogeneous catalyst in the preparation of thiazolylaryl α-aminophosphonates (**37**) (Scheme 30) [53].

6-Amino-1,3-dimethyluracil (**38**) was found to be a good starting material in condensation with benzaldehyde derivatives and diethyl phosphite. Phosphorus pentoxide in methanesulfonic acid (1:10), known as Eaton’s reagent, was the catalyst (Scheme 31) [54].

The condensation of the basic model (aniline derivatives, substituted benzaldehydes, and diethyl phosphite) was also performed with water as the medium, using different organic acid catalysts (Scheme 32) [55]. The yields were variable.

Potassium hydrogen sulfate also proved to be a powerful catalyst in the above type of reaction (Scheme 33) [56].

2-Cyclopropylpyridimidine-4-carbaldehyde (**40**) was also used as the starting material in three-component condensations. In one case, phosphomolybdic acid was applied as the catalyst (Scheme 34) [57]; in another, camphor-derived thiourea organocatalysts (**42, 43**) were utilized (Scheme 35) [58]. The yields of products (**41**) were mostly high. 

Zeolite derivatives such as H-β zeolite and MCM-41 were applied as green catalysts in different kinds of three-component condensations under discussion (Scheme 36 and Scheme 37) [59,60]. In the second series, quinoline-4-carbaldehyde (**44**) was the oxo component.

Regarding the acid-catalyzed Kabachnik–Fields reactions, those applying water as the solvent or a biodegradable catalyst, or those that may be performed under solvent-free conditions, can be considered “green”. PTSA has been used in various Kabachnik–Fields reactions as an efficient catalyst. The application of other acidic catalysts (camphor-derived thiourea catalysts, H-β zeolite, MCM-41) gave the corresponding products in variable yields.

The phospha-Mannich reaction of 5-hydroxymethyl-furan-1-carbaldehyde (**46**) with aniline and diethyl phosphite was catalyzed by elemental iodine, allowing the condensation under mild conditions (Scheme 38) [61]. It is noteworthy that 2-methyltetrahydrofuran was used as a green solvent. The yields of the α-aminophosphonates (**47**) were variable.

It was proven that the reaction proceeds via the imine pathway, which is followed by the nucleophilic attack of the diethyl phosphite to furnish the α-aminophosphonate. The role of iodine is to activate the imine(s) in the nucleophilic addition. Iodine may act as a Lewis acid [61].

α-(Furfurylamino)-alkylphosphonates (**49**) were synthesized by the Kabachnik– Fields reaction of furfurylamine (**48**), aromatic aldehydes, and dialkyl phosphites under MW irradiation. Silica-gel-supported iodine was used as the catalyst under solvent-free conditions. The plant growth regulatory activity of the products was investigated (Scheme 39) [62].

A Lewis-acid-type ionic liquid ([bmim][AlCl_4_]) was a novel catalyst in the three-component condensations performed under sonochemical irradiation (Scheme 40) [63].

### 2.3. Catalyst-Free Kabachnik–Fields Reactions

Having discussed the catalyst-promoted Kabachnik–Fields condensations, let us focus instead on the catalyst-free variations, which represent an environmentally friendly approach. The application of ionic liquids (**50** or **51**) as the solvent allowed the condensation of the three-components at room temperature (Scheme 41(1) and (2)) [64]. Both mono- (**4**) and bis products (**52**) were identified. The yields were variable, and fell in the range of 25–96%. There were no data provided on the recycling of the ionic liquids.

The expected reaction of alkylamines, 2-hydroxy-4-methoxyacetophenone (**53**), and a series of phosphites in boiling toluene followed an intramolecular cyclization to furnish heterocyclic α-aminophosphonates (**54**) (Scheme 42) [65].

In a somewhat analogous reaction, Bálint et al. reacted 2-formylbenzoic acid, primary amines, and dialkyl phosphites to afford, eventually, after an intramolecular cyclization, isoindolin-1-one-3-phosphonates [66]. 

The catalyst-free conversion of pyrene-1-carboxaldehyde (**56**) to the corresponding α-aminophosphonate (**57**) in reaction with amines (**55**) and dibenzyl phosphite is noteworthy (Scheme 43) [67]. It was observed that the corresponding α hydroxyphosphonate (**58**) was also present in the mixture formed as an intermediate in the Pudovik reaction.

A series of new α-aminophosphonates (**58**) was prepared applying aryl- or heteroarylaldehydes, aryl- or heteroarylamines, and diphenyl phosphite in three-component condensations using polyethylene glycol (PEG) as the solvent (Scheme 44) [68].

A phenothiazine carbaldehyde (**59**) was converted to different α-aminophosphonates (**60**) in a similar way (Scheme 45) [69].

Glycerol could also be used as the solvent in the catalyst-free condensation of amines, arylaldehydes, and phosphites (Scheme 46) [70].

A series of new α-sulfamidophosphonates and cyclosulfamidophosphonates incorporating quinoline or a quinolone moiety was synthesized by the Kabachnik–Fields reaction in the presence of a suitable ionic liquid under ultrasound irradiation [71].

A series of catalyst-free and MW-assisted Kabachnik–Fields reactions were elaborated by the Keglevich group. The first observation was that MW irradiation may substitute for the catalysts [17,72]. Among other examples, phospha-Mannich reactions with paraformaldehyde and ethyl phenyl-*H*-phosphinate were elaborated using primary and secondary amines (Scheme 47(1) and (2), respectively). Moreover, bis derivatives (**63**) were also prepared (Scheme 47(3)) [73]. The yields were variable.

A series of α-aminophosphonates (**64**, **66**, **68** and **70**) with sterically demanding α-aryl substituents was synthesized using a MW-assisted catalyst-free and solvent-free protocol (Scheme 48) [74].

A similar method was applied to the preparation of 6-methyl-2*H*-pyran-2-on (**71**)-based α-aminophosphonates and an α-aminophosphine oxide (**72**) (Scheme 49) [75].

Functionalized amines may also be used in the Kabachnik–Fields reaction. Amines with hydroxyalkyl substituents were condensed with paraformaldehyde and dialkyl phosphites or ethyl phenyl-*H*-phosphinate under MW irradiation (Scheme 50(1)). After measuring in the >P(O)H reagents and formaldehyde in a double amount, bis derivatives (**74**) were formed (Scheme 50(2)) [76].

The esters of glycine were converted to the corresponding bis(phosphonoylmethyl)amines (**75**) by MW-promoted reaction with two equivalents of paraformaldehyde and dialkyl phosphites (Scheme 51(1)). The application of diphenylphosphine oxide resulted in bis(phosphinoylmethyl)amine derivatives (**76**) (Scheme 51(2)) [77].

In a similar way, β-aminophosphonic derivatives (**77**) also served as starting materials for the respective bis(phosphonoylmethyl)amines (Scheme 52) [78].

A series of amines was utilized in a tandem Kabachnik–Fields reaction to afford the corresponding nonsymmetric bis(phosphinoylmethyl)amines (**79**) after a two-step transformation (Scheme 53(1)). Subsequently, after debenzylation of intermediate **79** (R = Bn) to species **80** (the details were not reported), valuable tris derivatives (**81**) were prepared (Scheme 53(2)) [79].

A dialkyl phosphite with an octyl and ethyl group was also utilized in the phospha-Mannich reaction, to give species **82** and the bis variation (**83**) (Scheme 54(1) and (2)) [80].

The results of the Keglevich group were summarized as conference proceedings [81]. The bis(phosphinoylmethyl)amine derivatives (**84**) were excellent precursors of the corresponding bisphosphines (**85**) and could be converted to ring Pt complexes (**86**) (Scheme 55) [82,83,84].

Optically active α-phenylethylamine (**87/1**) was applied in the preparation of chiral α-aminophosphonates (**88**, Y^1^ and Y^2^ = alkoxy), an α-aminophosphine oxide (**88**, Y^1^ and Y^2^ = Ph), and an α-aminophosphinate (**88**, Y^1^ = EtO Y^2^ = Ph), along with the corresponding bis derivatives (**89**) utilized in the synthesis of the corresponding ring Pt complex (**90**) (Scheme 56) [85].

α-Aminophosphines may be special ligands in platinum, palladium, and rhodium complexes [86].

It was rather surprising that carboxylic acid amides could also be utilized as starting materials in the Kabachnik–Fields reaction. However, the amides had to be applied in a 10-fold excess under practically solvolytic and forcing conditions (Scheme 57) [87].

Another series of bis(phosphine oxides) (**84**) was prepared via the double Kabachnik–Fields reaction (Scheme 58(1)). The precursors were converted to ring Pt complexes (**93**) (Scheme 58(2)). A few bisphosphines (**92**) were stabilized as bis(phosphine boranes) (**94**) (Scheme 58(3)). The catalytic activity of the Pt(II) complexes **93** was investigated in the hydroformylation of styrene. The advantages of applying Pt(II) complexes (**93**) include their high chemo- and regioselectivity at low temperatures [88].

α-Hydroxyphosphonates formed reversibly from suitable ketones and dialkyl phosphites [89] may also be converted to α-aminophosphonates by substitution reaction with amines. This reaction is enhanced by an adjacent group effect [90,91].

Somewhat analogous compounds, (aminomethylene)bisphosphine oxides and (aminomethylene)bisphosphonates (**95**/**96**), were prepared via the three-component condensation of secondary or primary amines, triethyl orthoformate, and the corresponding >P(O)H reagent (Scheme 59(1) and (2)) [92].

In the third part of the first section, green methods utilizing MW irradiation were summarized. These solvent- and catalyst-free protocols produce α-aminophosphonates, α-aminophosphinates, and phosphine oxides, along with their bis and tris derivatives in good yields. The advantages of applying MW irradiation include the mild reaction conditions, selectivity, and high yields. To compare the methods described, it goes without saying that MW-assisted accomplishment is the most suitable method to synthesize α-aminophosphonates and their derivatives.

### 2.4. Kabachnik–Fields Reactions Leading to Optically Active α-Aminophosphonates

Optically active α-aminophosphine oxides (**98**) were synthesized from the ethyl ester of proline (**97**), benzaldehyde derivatives, and diphenylphosphine oxide in toluene at reflux (Scheme 60) [93]. The chiral center in the proline derivative influenced the enantioselectivity. 

Starting from the optically active forms of α-phenylethylamines (**87/2**), benzaldehyde, and dimethyl phosphite, the corresponding products (**99/1, 99/2**) were formed in a diastereoselective manner under MW-assisted solvent-free and catalyst-free conditions (Scheme 61) [94].

In the above examples, the application of optically active amines as the starting materials allowed a diastereoselectivity of 74–92%.

## 3. Kabachnik–Fields Reactions Applying Trialkyl Phosphites or Related Derivatives as the P Reagent

In the next section, Kabachnik–Fields condensations applying trialkyl phosphites and related derivatives are discussed. Primary amines, 4-(4′-pyridyl)benzaldehyde (**100**), and triethyl phosphite were condensed in the presence of PEG–SO_3_H in toluene at 40–50 °C to give the corresponding aminophosphonates (**101**) (Scheme 62) [95].

The reaction of substituted anilines, benzaldehyde derivatives, and triethyl phosphite was carried out in the presence of the HCl salt of DABCO at 26 °C in MeOH (Scheme 63) [96].

Aniline derivatives and benzaldehyde, or aniline and benzaldehyde derivatives, were reacted in combination with triethyl phosphite in the presence of the T3P^®^ reagent at 26° C in ethyl acetate. It is a disadvantage that 1 equivalent of the reagent is needed in these condensations. The products (**4**) were obtained in 80–96% yields (Scheme 64(1) and (2)) [97].

A possible mechanism is demonstrated via the reaction of benzaldehyde, aniline, and triethyl phosphite (Scheme 65). In the first stage, the corresponding imine (**102**) is formed along with P, P′, P″-tripropyl triphosphonic acid (“T3P·H_2_O”) as the byproduct. The imine then reacts with triethyl phosphite, and, after protonation with “T3P·H_2_O”, the phosphonium salt (**104**) so formed is converted to the final α-aminophosphonate (**4** (R^3^ = Et)) via an Arbuzov fission. In this step T3P·EtOH is the byproduct [97].

Although the triesters of phosphorous acid are hydrolyzable in water, still water was used as the medium in a few cases. The interaction of primary amines, salicylaldehydes, and triphenyl phosphite using *p*-toluenesulfonic acid (PTSA) as the catalyst in water at room temperature afforded the corresponding aminophosphonates (**105**) in good yields (Scheme 66) [98].

The condensation of aniline and benzaldehyde with triethyl phosphite was also performed in water, applying the salts of dodecyl sulfonic acid or dodecylbenzene sulfonic acid (Scheme 67) [99].

Hafnium(IV) chloride was found to be an efficient catalyst in the condensation of amines/diamines, aldehydes, and trialkyl phosphites using ethanol as the solvent at 60 °C (Scheme 68) [100]. 

The following example for condensation in water involves the reaction of aniline and benzaldehyde derivatives with triethyl phosphite, utilizing a SO_3_H-functionalized ionic liquid as the catalyst (Scheme 69) [101].

The three-component reactions were also realized under ultrasonic irradiation at 26 °C in an ethyl lactate–water mixture (Scheme 70) [102]. Under these conditions, the corresponding products were obtained in good yields.

Ultrasound activation allowed a solvent- and catalyst-free condensation of aniline and benzaldehyde derivatives with triethyl phosphite (Scheme 71) [103].

We now consider further solvent-free methods. [Emim][Br] was found to be an efficient catalyst in the neat condensation of different amines, aldehydes, and phosphites, allowing the use of temperatures as low as 26 °C (Scheme 72) [104].

A solvent-free sonochemical transformation utilized a magnetically recoverable composite catalyst (Scheme 73) [105].

Boric acid was a suitable catalyst for the condensation of amines, benzaldehyde derivatives, and trimethyl phosphite (Scheme 74) [106].

Dicationic ionic liquids were used as recyclable catalysts in the solvent-free synthesis of aminophosphonates starting from primary amines, benzaldehyde derivatives, and trimethyl phosphite (Scheme 75) [107].

When an ortho-ethoxycarbonylmethyl-benzaldehyde derivative (**107**) was the oxo component, the condensation was followed by an intramolecular cyclization to furnish the corresponding isoquinolone derivative (**108**) (Scheme 76) [108].

A few other Kabachnik–Fields reactions followed by intramolecular cyclization have also been described. Ordóñez et al. elaborated the MW-assisted condensation of 2-formylbenzoic acid, aromatic amines (including optically active ones), and dimethyl phosphite [109,110,111] or triethyl phosphite [112] to afford the corresponding isoindolin-1-one-phosphonates after a ring closure in the final step.

To compare the methods with each other, a few studies have been performed in solvents, like the protocol using T3P, which gives the products in good yields. On the other hand, a few other methods applying Fe_3_O_4_@SiO_2_-imid PMA and H_3_BO_3_ as the catalysts in a solvent-free manner have been to give the corresponding products in good yields. The advantage of using dicationic ionic liquids is that the catalysts are recyclable. These issues are of importance from the point of view of green chemistry. These methods play an important role in allowing mild conditions, reducing the reaction times, and giving the products in high yields. However, if the Kabachnik–Fields reactions utilizing dialkyl phosphites and trialkyl phosphites are compared, the protocol applying dialkyl phosphites (see subchapter 3) is unambiguously the method of choice due to its atomic efficiency. Moreover, trialkyl phosphites have an unpleasant smell.

## 4. The aza-Pudovik Reaction

The aza-Pudovik reaction involving the addition of a dialkyl phosphite to an imine is another approach to synthesize α-aminophosphonates. A carbazole-related imine (**109**) was reacted with dialkyl phosphites and diphenylphosphine oxide in the presence of tetramethylguanidine as the catalyst in toluene to give the corresponding adducts (**110**) in good yields (Scheme 77) [113].

In another case, the simple starting materials benzylideneimines were reacted with secondary phosphine oxides, where a guanidium salt (**112**) served as the catalyst (Scheme 78) [114].

A cinchona-derived thiourea (**117**–**122**) was applied in the addition of diphenyl phosphite to ketimines (Scheme 79) [115]. The yields of products **114** and **116** varied within the range of 48–88%.

The aza-Pudovik approach was also useful in the synthesis of (5-nitrofuranoyl)-substituted esters of a phosphonoglycine derivative (**124**). In these cases, BF_3_·OEt_2_ was the catalyst (Scheme 80) [116,117].

This method was applied to the preparation of the corresponding pyrrole derivatives (**127**). In this case, the imine (**126**) was synthesized in situ (Scheme 81) [118].

In a variation, the imines generated in the reaction of aniline and benzaldehyde derivatives were reacted with diethyl phosphite in boiling ethanol to furnish the corresponding α-aminophosphonates (**129**) in medium yields (Scheme 82) [119].

The imines (**130**) obtained from benzaldehyde were reacted with diethyl phosphite in the presence of MoO_2_Cl_2_ as the catalyst under solvent-free conditions. (Scheme 83) [120].

α-Aryl-α-aminophosphonates and α-aryl-α-aminophosphine oxides (**132**) were prepared by the MW-assisted Pudovik reaction of benzylideneimines and >P(O)H reagents (Scheme 84) [121].

Poly-(4-vinylbenzaldehyde) (**134**) was prepared by the polymerization of 4-vinylbenzaldehyde (**133**) in DMSO. Postpolymerization modification reactions comprised imine formation (which is not shown here), and the addition of dialkyl phosphite to afford functional polymers. The α-aminophosphonate (**135**) scaffold was a side group (Scheme 85) [122].

In another approach, poly(aminophosphonates) were prepared in a one-pot manner including Kabachnik–Fields condensation and polymerization [123]. Functional polymers with bis(phosphonomethyl)amine moieties were also prepared. In this case, phosphorous acid was the P component [124].

Theoretical calculations predicted that the aza-Pudovik reaction under discussion takes place in a single concerted step involving transition state (**136**) formed from the trivalent tautomeric form of dimethyl phosphite and *N*-benzylideneaniline (Scheme 86) [121].

It was shown that the aza-Pudovik reaction involving the addition of >P(O)H reagents into the unsaturation of imines is a good and atomically efficient route for the preparation of α-aminophosphonic derivatives. As a matter of fact, as was shown in the Introduction, imines are intermediates of the Kabachnik–Fields reactions that may be formed from the oxo component and the amine.

## 5. Conclusions

In conclusion, various methods for the synthesis of α-aminophosphonic derivatives utilizing the Kabachnik–Fields reaction are summarized herein. In this review, different approaches utilizing a wide range of catalysts are summarized, encompassing the results of the last decade. We focused on environmentally friendly points of view. The solvent-free MW-assisted methods are of special importance. The Kabachnik–Fields reaction is also suitable for the synthesis of bis derivatives. Beside >P(O)H reagents, trialkyl phosphites may also be used as the starting materials of the phospha-Mannich condensation. The aza-Pudovik reaction is a special variation for the preparation of α-aminophosphonic derivatives and related compounds.

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
