# Peer review of "Synthesis of α-Aminophosphonates and Related Derivatives; The Last Decade of the Kabachnik–Fields Reaction"

_molecules, 2021, doi:10.3390/molecules26092511_

Round 1
Reviewer 1 Report
In this manuscript, Keglevich and coworker reported in a review the last ten years of the Kabachnik–Fields Reaction for the synthesis of α-aminophosphonates and Related Derivatives which has resulted to be a very prolific area in organic chemistry.
Some general (some specific) comments and suggestions about the manuscript are presented below:
- During the hole manuscript the authors refers to green chemistry and the hole manuscript’s body is directed to green chemistry. Then, I would suggest to comment something about that in the introduction and also introduce the fact that this green chemistry precepts has serve to organize the manuscript.
- In the introduction, I would like to see a brief scheme about the general mechanism expected for the Kabachnik–Fields Reaction.
- If possible, I would add subsections (at least in section 2) in order to make it simpler for the reader. Then, for example in section 2, I would add subsections with metal-catalysed systems, acid-catalysed systems, etc. Following with previous point, I have observed that when changing to new family of conditions or promoters a kind of conclusion can be read. this will have much more sense if a subsection could be clearly identified.
- I think that would be very interesting to see in a scheme or read in the discussion the role (mechanisticly) of the promoters presented during the manuscript (at least one for each family or promoters), particularly for the cases of non-common promoters such for example I2 or T3P. I have found slightly poor the discussion about the examples showed in the schemes. This would help for example for Scheme 14. This reaction is included in acidic catalyst. However, the MOF used in the reaction I do not know if can be clearly identified as an acid.
- As general comment for the hole manuscript, I would simplify the schemes removing all the enormous and confusing tables. I would just add the required substituents of the substrates in a simplified mode within the scheme (Ar, Alkyl, etc. in a general mode. No need of add every single substituent).
- I have seen very interesting asymmetric examples. I would create a new section for those, adding, of course, the information about the enantiocontrol observed for each reaction.
- Scheme 55, which has been included in the block where trialkylphosphates are included, do not describe alkyl ones.
- “However, if the Kabachnik–Fields reactions utilizing dialkyl phosphites and trialkyl phosphites are compared, unambiguously the protocol applying dialkyl phosphites is the method of choice due to the atomic efficiency.” During the previous examples showed before that sentence, I could only find a couple of examples about dialkyl phosphites applied in a Kabachnik–Fields reaction. I think that this information could have been discussed in the specific examples.
- Again, I would show a general mechanism in section 4.
In my opinion, the presented manuscript shows interesting information about an apparently widely explored reaction. I think that I would recommend it to be published in Molecules after addressing the above-mentioned comments. In addition, I would recommend the authors to check the English (language and style) of the entire manuscript.
Author Response
Referee 1
The au-s are grateful for the Referee for the very good, indeed bettering suggestions.
“Some general (some specific) comments and suggestions about the manuscript are presented below:
- During the whole manuscript the authors refers to green chemistry and the hole manuscript’s body is directed to green chemistry. Then, I would suggest to comment something about that in the introduction and also introduce the fact that this green chemistry precepts has serve to organize the manuscript.”
This request was obeyed in the Introduction. See the sentence on p. 2 l. 49-50.
“- In the introduction, I would like to see a brief scheme about the general mechanism expected for the Kabachnik–Fields Reaction.”
This wish was realized, see new Scheme 1.
“- If possible, I would add subsections (at least in section 2) in order to make it simpler for the reader. Then, for example in section 2, I would add subsections with metal-catalysed systems, acid-catalysed systems, etc. Following with previous point, I have observed that when changing to new family of conditions or promoters a kind of conclusion can be read. this will have much more sense if a subsection could be clearly identified.”
New subtitles were inserted as follows:
2.1 Metal catalyzed K-F reactins
2.2 Acid-catalyzed K-F reactions
2.3 Catalyst-free K-F reactions
2.4 K-F reactions leading to optically active alfa-aminophosphonates
“- I think that would be very interesting to see in a scheme or read in the discussion the role (mechanisticly) of the promoters presented during the manuscript (at least one for each family or promoters), particularly for the cases of non-common promoters such for example I2 or T3P.”
This advice was well realized. See new Scheme 1 (mentioned above), the new text inserted after the I2-catalyzed condensation (below Scheme 38), the detailed mechanism inserted for the T3P-promoted reactions using triethyl phosphite (see new Scheme 65 + new text), and the mechanism proposed for the aza-Pudovik reactions (see new Scheme 86 + new text).
“I have found slightly poor the discussion about the examples showed in the schemes. This would help for example for Scheme 14. This reaction is included in acidic catalyst. However, the MOF used in the reaction I do not know if can be clearly identified as an acid.”
In the revised version this is Scheme 19. Well, above the scheme it was clearly written that the acidic component of the not simple catalyst is 2-aminoterephthalic acid
“- As general comment for the hole manuscript, I would simplify the schemes removing all the enormous and confusing tables. I would just add the required substituents of the substrates in a simplified mode within the scheme (Ar, Alkyl, etc. in a general mode. No need of add every single substituent).”
We ask the understanding of the Referee that this was a huge work to collect all combinations of the substituents. In our opinion, this is a serious added value. If the possible substituents were simply listed, the combinations would be lost. However, we tried to simplify the tables removing the horizontal lines.
“- I have seen very interesting asymmetric examples. I would create a new section for those, adding, of course, the information about the enantiocontrol observed for each reaction.”
As it was mentioned above, the examples starting from optically active derivatives were grouped into new sub-chapter 2.4.
“- Scheme 55, which has been included in the block where trialkylphosphates are included, do not describe alkyl ones.”
Of course, subtitle 3 was corrected to “K-F reactions applying trialkyl phosphites or related derivatives as the P-reagent”
- “However, if the Kabachnik–Fields reactions utilizing dialkyl phosphites and trialkyl phosphites are compared, unambiguously the protocol applying dialkyl phosphites is the method of choice due to the atomic efficiency.” During the previous examples showed before that sentence, I could only find a couple of examples about dialkyl phosphites applied in a Kabachnik–Fields reaction. I think that this information could have been discussed in the specific examples.”
This may be a misunderstanding, s the whole chapter 2 is on the use of dialkyl phosphites or related derivatives, e. g. (PhO)2P(O)H. The final comment is valid for all examples.
“- Again, I would show a general mechanism in section 4.”
As it was mentioned above, a general mechanism was included at the end of the main part.
“In addition, I would recommend the authors to check the English (language and style) of the entire manuscript.”
This was checked carefully and we made not less amendments.
All changes were highlighted in yellow.
Reviewer 2 Report
The manuscript is intended to summarize the major developments of the Kabachnik-Fields reaction and covers the literature published after the first review by the Authors (Molecules 2012, 17, 12821). Since the topic still attracts considerable interest, that is illustrated by over 450 papers appeared after 2012, the present review is purposeful and expected. Several important aspects of synthetic procedures leading to α-aminophosphonates, α-aminophosphinates, and α-aminophosphine oxides were covered including metal-catalyzed processes and also discussing the use of organocatalysts, metal-organic framework-based catalysts and reactions in ionic liquids. Several examples of solvent- and catalyst-free protocols were also mentioned. In summary, the scope and completeness of the manuscript is adequate. The text is well-written with only few typographic errors. In my opinion, the presented manuscript deserves publication.
Minor points:
- page 2, line 60 aldehyde not aldehide
- Scheme 41, stereochemistry of the methyl group in compounds 69/1 and 69/2 should be shown
- In some cases the use of chiral catalyst resulted in stereoselective formation of the product (ref. 97 and 98). Therefore, the effectiveness of the chirality transfer should be mentioned, possibly along with the visualisation of the stereochemistry of final compounds.
Author Response
Referee 2
First of all, the au-s express their thanks to this Referee for her/his positive opinion.
Minor points:
- “page 2, line 60 aldehyde not aldehide”
This was corrected.
- “Scheme 41, stereochemistry of the methyl group in compounds 69/1 and 69/2 should be shown”
The scheme (in the revised version Scheme 61) was corrected.
- “In some cases the use of chiral catalyst resulted in stereoselective formation of the product (ref. 97 and 98). Therefore, the effectiveness of the chirality transfer should be mentioned, possibly along with the visualisation of the stereochemistry of final compounds.”
The chirality transfer was shown and mentioned in the text in Schemes 78 and 79.
All changes were highlighted in yellow.
Reviewer 3 Report
The last review on the synthesis of α-aminophosphonates through the Kabachnik-Fields reaction by Prof. Keglevich in 2012 and not all the publications of this year were included, I consider that in this last review should also include the publications of 2012 to which It goes from 2021 to fulfill what the authors mention, the review of the last decade (ten yeras).
I request that the authors make a more careful review, because several publications have not been included, for example:
ACS Omega 2021, 6, 2934-2948
ACS Macro Lett. 2014, 3, 4, 329–332
Synlett 2014, 25, 1145-1149
Arch. Pharm 2021, 354. E2000291
ChemistrySelect 2020, 5, 13454-13460
Res. Chem. Intermediates 2021, 47, 1139-1160
Polym. Chem., 2014, 5, 1857
Macromol. Rapid Commun. 2015, 36, 828−833
Bioorg. Med. Chem. Lett. 26 (2016) 1310–1313
Bioorganic Chemistry 69 (2016) 132–139
Tetrahedron Letters 57 (2016) 1782–1785
Tetrahedron Letters 53 (2012) 5497–5502
Synlett 2012, 23, 1931-1936
Journal of Molecular Structure 1134 (2017) 217-225
Tetrahedron Letters 54 (2013) 6403–6406
Tetrahedron 74 (2018) 1817-1825
Also some reviews were not cited
Add enantiomeric and diastereoisomeric excesses when chiral substrates or catalysts are used.
There are some interesting examples about the Kabachnik-Fields reaction followed by an intramolecular cyclization, as in the Scheme 63 and they were not included.
Improve table presentation
Group all the reactions that are carried out by microwave or by catalyst type
If possible group trisubstituted and tetrasubstituted α-aminophosphonates as well as acyclic and cyclic.
Author Response
Referee 3
First, let we thank this Referee for his/her constructive and indeed bettering remaks. All suggestions were obeyed.
“The last review on the synthesis of α-aminophosphonates through the Kabachnik-Fields reaction by Prof. Keglevich in 2012 and not all the publications of this year were included, I consider that in this last review should also include the publications of 2012 to which It goes from 2021 to fulfill what the authors mention, the review of the last decade (ten years).
I request that the authors make a more careful review, because several publications have not been included, for example:
ACS Omega 2021, 6, 2934-2948
ACS Macro Lett. 2014, 3, 4, 329–332
Synlett 2014, 25, 1145-1149
Arch. Pharm 2021, 354. E2000291
ChemistrySelect 2020, 5, 13454-13460
Res. Chem. Intermediates 2021, 47, 1139-1160
Polym. Chem., 2014, 5, 1857
Macromol. Rapid Commun. 2015, 36, 828−833
Bioorg. Med. Chem. Lett. 26 (2016) 1310–1313
Bioorganic Chemistry 69 (2016) 132–139
Tetrahedron Letters 57 (2016) 1782–1785
Tetrahedron Letters 53 (2012) 5497–5502
Synlett 2012, 23, 1931-1936
Journal of Molecular Structure 1134 (2017) 217-225
Tetrahedron Letters 54 (2013) 6403–6406
Tetrahedron 74 (2018) 1817-1825”
O yes, the Referee is right, we overlooked the above articles obviously relevant to this new review paper. The literature survey was not complete. Sorry for this. We have included all the 16 suggested references. In almost all cases new schemes were also drawn supplied with the new text. See new Schemes 3, 12, 17, 18, 21, 22, 23, 30, 39, 68, 71, 82 and 85. In a few cases [71, 124 and 124], we only texually included the information covered by the references.
“Also some reviews were not cited.”
We did our best to cite all relevant review articles.
“Add enantiomeric and diastereoisomeric excesses when chiral substrates or catalysts are used.”
This was done, where it was relevant. See e. g. Schemes 60,61, 78 and 79.
“There are some interesting examples about the Kabachnik-Fields reaction followed by an intramolecular cyclization, as in the Scheme 63 and they were not included.”
Similar follow-up cyclization reactions [109-111] were mentioned after Scheme 76 (in the revised version), and also after Scheme 42 [66] (in the earlier part). All together 5 additional references were included re cyclizations.
“Improve table presentation.”
We tried to make more esthetic the tables of substituents by removing the horizontal lines.
“Group all the reactions that are carried out by microwave or by catalyst type.”
In whole new subtitles were inserted (see below), and within them – if there were – the catalyst were grouped in a more clear-cut way. However, as the MW examples occurred rather “heterogeneously”, we could not group together these cases.
2.1 Metal catalyzed K-F reactins
2.2 Acid-catalyzed K-F reactions
2.3 Catalyst-free K-F reactions
2.4 K-F reactions leading to optically active alfa-aminophosphonates
If possible group trisubstituted and tetrasubstituted α-aminophosphonates as well as acyclic and cyclic.
This could have caused a difficulty for us in the discussion, but as it was mentioned under the previous point, another kind of grouping has been perfectuated. Hope this is acceptable for the Referee.
All changes along with the new schemes were highlighted in yellow.
Round 2
Reviewer 3 Report
Once some other articles have been added, and some modifications made to the tables, this manuscript can be accepted for publication.